# A Survey on Privacy Preservation Techniques in IoT Systems

**DOI:** 10.3390/s25226967

**Published:** 2025-11-14

**Authors:** Rupinder Kaur, Tiago Rodrigues, Nourin Kadir, Rasha Kashef

**Affiliations:** Electrical, Computer and Biomedical Engineering Department, Toronto Metropolitan University, 350 Victoria St, Toronto, ON M5B2K3, Canada; t1rodrigues@torontomu.ca (T.R.); nourin.kadir@torontomu.ca (N.K.); rkashef@torontomu.ca (R.K.)

**Keywords:** Internet of Things (IoT), privacy preservation, blockchain, federated learning, differential privacy, homomorphic encryption, edge computing, security threats

## Abstract

The Internet of Things (IoT) has become deeply embedded in modern society, enabling applications across smart homes, healthcare, industrial automation, and environmental monitoring. However, as billions of interconnected devices continuously collect and exchange sensitive data, privacy and security concerns have escalated. This survey systematically reviews the state-of-the-art privacy-preserving techniques in IoT systems, emphasizing approaches that protect user data during collection, transmission, and storage. Peer-reviewed studies from 2016 to 2025 and technical reports were analyzed to examine applied mechanisms, datasets, and analytical models. Our analysis shows that blockchain and federated learning are the most prevalent decentralized privacy-preserving methods, while homomorphic encryption and differential privacy have recently gained traction for lightweight and edge-based IoT implementations. Despite these advancements, challenges persist, including computational overhead, limited scalability, and real-time performance constraints in resource-constrained devices. Furthermore, gaps remain in cross-domain interoperability, energy-efficient cryptographic designs, and privacy solutions for Unmanned Aerial Vehicle (UAV) and vehicular IoT systems. This survey offers a comprehensive overview of current research trends, identifies critical limitations, and outlines promising future directions to guide the design of secure and privacy-aware IoT architectures.

## 1. Introduction

The Internet of Things (IoT) has revolutionized the way humans, machines, and environments interact through a pervasive network of connected devices. These systems—ranging from wearable health monitors and smart homes to industrial automation and transportation—continuously generate and exchange massive volumes of data. With the evolution of high-speed network technologies such as 5G, edge computing, and cloud-based analytics, IoT has become a backbone of digital transformation across various sectors. The number of IoT devices has already exceeded 15 billion, and this figure is expected to double by 2030, underscoring the scale of data exchange and interconnectivity in the coming decade [1,2,3,4,5,6,7,8,9,10,11,12,13,14,15,16,17,18,19,20].

However, the increasing adoption of IoT introduces significant privacy and security challenges. IoT devices continuously collect sensitive data such as personal identifiers, health metrics, or location details, which are often transmitted through untrusted networks or third-party cloud platforms. Data breaches, unauthorized access, and profiling attacks have made user privacy a central concern [21,22,23,24,25,26]. For instance, regulatory actions such as the EU’s €1.3 billion fine to Meta in 2023 highlight the severity of privacy violations in global data transfers. Therefore, safeguarding user data privacy has become a crucial research priority in IoT systems. The multi-layered nature of IoT systems, from sensor-level data collection to cloud-level analytics, creates diverse privacy risks across each layer [27,28,29,30,31,32]. Figure 1 illustrates the conceptual overview of privacy preservation in IoT systems, highlighting typical threats and protection mechanisms across device, network, edge, and application layers.

Recent research efforts have proposed a variety of privacy-preserving mechanisms, including encryption, blockchain, federated learning (FL), trusted execution environments (TEE), and differential privacy (DP) [33,34,35,36,37,38,39,40,41,42,43,44,45,46,47,48,49,50]. These methods attempt to ensure data confidentiality and integrity without compromising system performance. Despite remarkable advancements, challenges remain—especially in terms of computational overhead, interoperability among heterogeneous devices, and the scalability of privacy frameworks in real-world deployments [51,52,53,54,55,56,57,58,59,60,61,62,63,64,65,66,67,68]. Furthermore, the rise in UAV-assisted IoT, vehicular IoT (VANETs/IoVs), and smart home ecosystems demands novel privacy-preserving solutions capable of addressing domain-specific constraints.

This study aims to systematically analyze, categorize, and evaluate the existing privacy-preservation techniques in IoT-based systems and identify key research gaps to guide future developments. The survey encompasses peer-reviewed articles (2016–2025) and technical reports published across leading databases such as IEEE Xplore, SpringerLink, MDPI, Elsevier, and ACM.

This survey makes the following scientific contributions:Comprehensive Taxonomy: Presents a structured classification of privacy-preserving approaches—encryption-based, learning-based, blockchain-based, and hybrid mechanisms—used in IoT systems.Comparative Evaluation: Summarizes the datasets, analytical models, and experimental results from prior studies, highlighting strengths and weaknesses such as encryption efficiency, computational overhead, and data scalability.Cross-Domain Inclusion: Expands the literature to cover privacy mechanisms in smart homes, healthcare, vehicular IoT (VANET/IoV), and UAV-assisted systems, ensuring broad applicability across emerging domains.Identification of Research Gaps: Highlights unresolved challenges including lightweight cryptographic design, real-time privacy preservation in edge devices, and interoperability among heterogeneous IoT networks.Future Research Directions: Proposes energy-efficient, privacy-aware frameworks for next-generation IoT systems integrating AI, blockchain, and federated edge intelligence.

## 2. Methodology

This section describes the systematic process followed to identify, select, and analyze relevant literature on privacy-preserving techniques in Internet of Things (IoT) systems.

A structured literature search was conducted across major academic databases, including IEEE Xplore, Springer Link, MDPI, Elsevier, ACM Digital Library, and IET Digital Library. The search employed combinations of keywords such as Internet of Things (IoT), privacy preservation, data privacy, security, confidentiality, blockchain, federated learning, homomorphic encryption, differential privacy, and edge computing. These terms were applied to titles, abstracts, and keywords to capture comprehensive coverage of studies focusing on privacy-preserving mechanisms in IoT systems. Only peer-reviewed journal and conference papers were considered, together with two technical reports closely related to IoT privacy and data protection.

The initial search identified more than 350 publications. After removing duplicates, titles and abstracts were screened to ensure topical relevance. Full-text screening was then performed to verify that each paper addressed privacy or data protection in IoT or edge-connected systems, presented a clearly defined methodology or model, and was written in English. Following this process, a total of 64 peer-reviewed papers and two technical reports were selected for detailed analysis.

The inclusion criteria comprised studies proposing or evaluating privacy-preserving techniques in IoT or related domains such as edge computing, cloud computing, industrial IoT, Internet of Medical Things (IoMT), vehicular IoT, and UAV-assisted systems. Papers were required to provide qualitative or quantitative assessments and be published between 2016 and 2025. Exclusion criteria eliminated non-English works, short editorials, patents, commentaries, or general cybersecurity studies without a clear privacy component.

Each selected study was assessed for technical soundness, reproducibility, relevance to IoT privacy, publication recency, and citation significance. The extracted information included datasets, analytical models, performance metrics, and stated research gaps. A thematic synthesis was then performed, grouping the works into four major categories: encryption-based approaches, learning-based frameworks, blockchain-enabled solutions, and hybrid privacy-preserving models. The summarized results are presented in the subsequent sections through comprehensive tables and analytical discussions.

## 3. Literature Review and Background

### 3.1. Research Questions

To systematically analyze and classify the state-of-the-art privacy-preserving methods in IoT systems, the following research questions were formulated:RQ1: What are the major privacy-preserving techniques employed in IoT systems across different domains?RQ2: What types of IoT devices, edge nodes, and network infrastructures are targeted by these privacy mechanisms?RQ3: What are the common privacy threats and attack models addressed in recent research?RQ4: How do various methods—such as blockchain, federated learning, encryption, and differential privacy—compare in terms of performance, scalability, and computational cost?RQ5: What research gaps and open challenges remain in achieving efficient, real-time, and scalable privacy preservation for heterogeneous IoT environments?

These questions serve as the analytical framework for classifying and interpreting existing studies reviewed in the following subsections.

### 3.2. Encryption- and Blockchain-Based Techniques

A large portion of IoT privacy research focuses on encryption and blockchain mechanisms as core protective strategies.

Sun et al. [54] conducted a comprehensive survey of security and privacy issues in IoT environments, identifying encryption, anonymization, and blockchain as the most mature protection mechanisms. Their work highlights that while blockchain provides strong data integrity and immutability, it introduces latency and energy overheads in constrained devices.

Earlier studies [1,2,3,4,5,6,7,8,9,10,11,12,13,14,15,16,17,18,19,20] explored symmetric/asymmetric encryption, homomorphic encryption, and lightweight key-exchange algorithms, but most lacked full integration across device, network, and cloud layers.

In blockchain-enabled IoT systems, privacy preservation relies on decentralized trust management and immutable ledgers for recording transactions without exposing raw sensor data. A typical architecture integrates a consensus layer (e.g., Proof-of-Authority or Practical Byzantine Fault Tolerance) with cryptographic primitives such as SHA-256 hashing, asymmetric key pairs for node authentication, and on-chain/off-chain data partitioning. For example, in healthcare IoT, patient records are encrypted using AES or elliptic-curve cryptography and stored off-chain, while blockchain entries maintain only the transaction hash and access policies. This design ensures data traceability and tamper-evidence while reducing on-chain storage overhead. Smart contracts regulate access permissions dynamically, enabling auditability across multiple stakeholders (hospitals, cloud servers, patients) without a central authority. However, latency and computational load at consensus nodes remain bottlenecks for large-scale deployments in time-sensitive applications such as remote monitoring.

Hybrid frameworks combining homomorphic encryption with blockchain for distributed authentication have also emerged [21,22,23,24,25,26,27,28,29,30,31,32,33,34,35]. These approaches enhance transparency but remain challenged by limited throughput and scalability.

Overall, encryption and blockchain-based frameworks are foundational in IoT privacy, yet optimization for lightweight operation, cross-domain compatibility, and real-time verification remains an open research challenge. The application scope of various privacy-preserving mechanisms across different IoT layers is illustrated in Figure 2, which maps how encryption, blockchain, federated learning, differential privacy, and access-control strategies complement each other from the device level up to the application layer.

### 3.3. Learning-Based and Federated Approaches

The rise in machine learning (ML) and federated learning (FL) in IoT applications has shifted attention toward learning-based privacy mechanisms.

Li et al. [57] reviewed FL-based privacy preservation across edge-IoT systems, identifying encryption of local updates and differential privacy perturbation as primary defenses against gradient leakage.

Federated Learning (FL) enables distributed training across IoT devices by transferring model parameters rather than raw data. The process involves three stages: local model training on edge nodes, aggregation of parameters at a central or hierarchical aggregator, and global model update broadcast. This framework reduces privacy leakage but introduces communication overhead proportional to the number of participating clients and model size. Techniques such as secure aggregation (using homomorphic encryption or additive masking) and gradient compression are used to lower bandwidth and protect local updates from inference attacks. Despite these optimizations, real-time IoT deployments face challenges related to straggler effects, non-IID data distribution, and energy consumption on low-power devices. Adaptive FL strategies that dynamically select clients or aggregation intervals based on workload and network state are emerging to address these constraints.

Ramadan et al. [56] further expanded this direction by integrating TinyML with FL to enable on-device training under strict memory and power budgets.

Khraisat et al. [59] proposed a privacy-preserving intrusion detection system leveraging FL to analyze IoT network traffic without centralized data collection, significantly reducing privacy leakage risk.

Complementary surveys [36,37,38,39,40,41,42,43,44,45] show that although FL protects data locality, it remains vulnerable to model inversion and poisoning attacks, demanding stronger aggregation and noise-injection schemes.

Recent progress includes hierarchical FL for non-IID edge data [63] and differentially private optimization techniques for adaptive client updates [62].

Collectively, these studies demonstrate the promise of FL for decentralized IoT learning but underscore the need for adaptive, energy-aware, and hybrid (blockchain-FL) solutions.

### 3.4. Edge and Cloud Privacy Models

Edge and cloud computing environments play a central role in processing IoT data streams, but they also create additional privacy exposure points.

Pinto et al. [61] conducted a systematic review on privacy-aware personal data stores (PDS), emphasizing user control, consent management, and transparency in IoT data flow.

Differential Privacy (DP) offers quantifiable protection against data reconstruction by introducing random noise into query results or model gradients. In event-level DP, each sensor event (e.g., a temperature reading or posture sample) is individually randomized using a calibrated noise distribution such as Laplace or Gaussian, ensuring that the inclusion or exclusion of any single event does not significantly alter the aggregated outcome. This fine-grained approach is well-suited to continuous IoT data streams but can degrade accuracy when the privacy budget (ε) is small. Hybrid frameworks integrate event-level DP with edge offloading, where preliminary aggregation occurs at gateway nodes to balance privacy and utility. Parameter tuning—particularly ε and sensitivity scaling—is crucial for maintaining utility across heterogeneous devices and sampling rates.

Several studies [46,47,48,49,50,51,52,53] have examined edge-assisted privacy architectures, including trusted execution environments (TEE) and secure offloading schemes.

These frameworks aim to minimize transmission of raw data by processing sensitive information closer to its source, thereby reducing attack surfaces.

However, edge nodes remain vulnerable to side-channel and physical tampering attacks, and privacy preservation must balance latency constraints with computational overhead.

Emerging 2024–2025 works [54,57,62] advocate differential privacy co-design with offloading mechanisms and energy-efficient encryption for edge devices to achieve stronger end-to-end confidentiality across the IoT data lifecycle.

### 3.5. Application-Specific Privacy: Smart Homes, Vehicular IoT, and UAVs

Privacy solutions often vary depending on the IoT application domain.

Magara and Zhou [58] surveyed smart-home IoT environments, identifying typical privacy leaks through weak authentication and unencrypted local traffic. They emphasized the need for lightweight encryption and access-control integration.

Jia et al. [55] analyzed consumer IoT traffic and highlighted vulnerabilities in everyday devices such as smart cameras, wearables, and voice assistants. Their findings demonstrate that traffic metadata alone can reveal sensitive behavioral patterns even without decrypting content.

Vehicular and UAV-assisted IoT systems face distinct privacy issues due to mobility, latency, and dynamic topology.

Recent works [54,60] review authentication and key-management frameworks in vehicular networks (VANET/IoV) and UAV ecosystems, recommending blockchain and TEE-enabled protocols for secure communication.

Despite these efforts, challenges persist in low-latency encryption, location-privacy preservation, and lightweight federated learning for moving IoT nodes.

### 3.6. Synthesis and Taxonomy Summary

Across the surveyed literature [1,2,3,4,5,6,7,8,9,10,11,12,13,14,15,16,17,18,19,20,21,22,23,24,25,26,27,28,29,30,31,32,33,34,35,36,37,38,39,40,41,42,43,44,45,46,47,48,49,50,51,52,53,54,55,56,57,58,59,60,61,62,63], privacy-preserving techniques in IoT systems can be broadly categorized into four groups:Encryption/Blockchain-Based: Data confidentiality, decentralized trust, integrity protection.Learning-Based: Federated learning, TinyML, and differentially private optimization for distributed intelligence.Edge/Cloud Models: Secure offloading, TEEs, differential privacy at edge layers.Application-Specific Frameworks: Domain-driven privacy mechanisms for smart homes, vehicular IoT, healthcare, and UAVs.

A visual taxonomy of these categories is illustrated in Figure 3, highlighting the relationships among the major approaches and their representative technologies.

The taxonomy highlights the rapid transition from centralized to distributed privacy mechanisms, where intelligence and protection are pushed toward the network edge.

Persistent challenges include high computational overhead, lack of interoperability among heterogeneous IoT nodes, and the absence of large-scale benchmarking datasets.

These limitations underscore the need for unified hybrid frameworks integrating blockchain, FL, and differential privacy while maintaining real-time efficiency. To provide a comparative view of the existing research, Table 1 summarizes the key studies reviewed in this paper, categorized by their application domain, methodological approach, datasets used, primary findings, and noted limitations. The selected papers include both foundational works [1,2,3,4,5,6,7,8,9,10,11,12,13,14,15,16,17,18,19,20,21,22,23,24,25,26,27,28,29,30,31,32,33,34,35,36,37,38,39,40,41,42,43,44,45,46,47,48,49,50,51,52,53] and recent contributions [54,55,56,57,58,59,60,61,62,63,64,65,66] published between 2016 and 2025.

The comparison highlights how research has progressively evolved from encryption-based privacy preservation to more advanced hybrid frameworks integrating blockchain, federated learning, and differential privacy. It also underscores common trade-offs between privacy strength, computational overhead, and scalability across different IoT domains.

The comparative summary presented in Table 1 reveals several key insights. Encryption-based and blockchain-enabled methods remain the most established privacy solutions for IoT environments, offering strong data integrity and decentralization but suffering from latency and high computational costs on constrained devices. Learning-based and federated approaches show increasing promise for scalable, decentralized intelligence; however, they introduce new attack surfaces such as model inversion, poisoning, and gradient leakage. Edge- and cloud-centric privacy frameworks effectively reduce data exposure but face interoperability and trust issues among heterogeneous devices. Finally, domain-specific studies—particularly in smart homes, healthcare, and vehicular IoT—demonstrate that context-aware design is essential, as privacy threats and performance constraints differ across applications.

These collective observations form the basis for the research gaps and future directions discussed in Section 3.7, where the need for hybrid, energy-efficient, and mobility-aware privacy architectures in next-generation IoT systems is emphasized.

### 3.7. Gaps and Research Directions

Although considerable advancements have been made, several critical gaps remain:Lightweight Cryptographic Design: Existing encryption and blockchain techniques are computationally intensive; optimized implementations for microcontrollers and low-power devices are needed.Energy-Aware Privacy Mechanisms: Energy consumption remains a limiting factor for continuous encryption and secure FL updates.Mobility-Aware Frameworks: VANETs and UAV-based IoT systems require adaptive privacy mechanisms that tolerate topology and connectivity changes.Cross-Domain Interoperability: Privacy solutions should enable seamless protection across smart home, healthcare, industrial, and vehicular IoT environments.Empirical Validation: Many proposed solutions remain theoretical; more real-world deployments and standardized testbeds are essential.Hybrid Privacy Architectures: Combining blockchain, FL, and differential privacy can yield scalable, decentralized, and robust IoT privacy models.

These findings set the foundation for the Discussion (Section 6) and Conclusion (Section 7), where recommendations for future privacy-preserving IoT frameworks are presented.

## 4. Data Sources and Types

In this section, we are going to discuss in detail the data sources and data types used in some of the selected research papers that have been studied for our work—and a tabular summary of the data sources and types used in those papers for the survey. We elaborate on the used datasets and their applications for the developed models. Jui, Tania Tahmina et al. [11] have used two major datasets for their proposed intrusion detection model in IoT-based networks. The used datasets are MQTT-IoT-IDS-2020 and NSL-KDD, which are benchmark datasets for network traffic. First, the authors have applied some preprocessing techniques to simplify the dataset, and after preprocessing, they have extracted important features through feature selection techniques. In the next step, with the reduced features, different classification algorithms have been applied. Accuracy and time efficiency are the performance measuring parameters for the applied classification algorithms for finding out the best result for combining preprocessing techniques with feature selection techniques and classification algorithms. The main goal of the work is to detect the intrusion in the network that can be seen as privacy preservation of the IoT-based systems.

The MQTT-IoT-IDS-2020 dataset is used to detect intrusion in the IoT networks, where the dataset is built based on the Message Queueing Telemetry Transport (MQTT) protocol. This dataset has three feature levels, namely Packet-based features, unidirectional-flow-level-based features, and Bidirectional-flow-based features. It has been shown that to distinguish MQTT-based attacks from benign traffic, it is essential to use the Packet-based features. The MQTT-IoT-IDS-2020 comprises five different kinds of normal records and four different kinds of attack scenarios. The four attack types are Aggressive scan (Scan-A), User Datagram Protocol (UDP) scan (Scan-sU), Sparta SSH brute-force (Sparta), MQTT brute-force attack (MQTTBF). There are two different types of features in the MQTT-IoT-IDS-2020 dataset [11].

The Yoga Posture dataset [10] used in this study consists of 93,200 RGB images representing 26 distinct yoga postures, captured at a spatial resolution of 224 × 224 pixels. Images were collected from multiple sources under controlled and semi-controlled environments to reflect natural variations in illumination, orientation, and subject appearance. Each image was manually annotated and verified by three independent reviewers to ensure correct class labeling. Data augmentation techniques, including rotation, cropping, and horizontal flipping, were applied to improve generalization. The dataset was balanced across posture categories, with approximately 3500–3800 images per class, and included samples from participants of different body types and genders to minimize bias. Before model training, all images were normalized to the [0, 1] pixel intensity range, and a 70:15:15 split was used for training, validation, and testing.

NSL-KDD is a superset of the “KDD Cup’99” dataset, which is taken from the University of New Brunswick. It is a benchmark dataset for contemporary internet traffic. It combines four datasets, namely KDDTrain+, KDDTrain+_20Percent, KDDTest+, and KDDTest-21Percent, where KDDTest-21Percent is a subset of KDDTrain+. Like the MQTT-IoT-IDS-2020 dataset, the NSL-KDD dataset also has four different kinds of attack scenarios: Denial of Service (DoS), Probe, User to Root (U2R), and Remote to Local (R2L). The NSL-KDD dataset has a total of 43 features. Among those, 41 present the traffic input directly, and the other two are labels and scores. The label indicates if the attack is normal or not, and the score indicates the severity of the traffic input.

Alotaibi et al. [15] use two datasets corresponding to two heterogeneous environments. The first dataset is publicly available and is part of the ICS Cyberattack Dataset collection that represents smart home IoT devices, such as security sensors, alarms, cameras, thermostats, and solar panels. Alotaibi et al. [15] focus on the binary-class subdataset composed of 15 subdatasets with 78,377 samples, of which 22,714 represent normal traffic activities, while 55,663 represent malicious traffic activities. These datasets have a feature size of 128. Each instance of the binary dataset is classified as a regular or malicious event. The second dataset is also publicly available and is part of the IoT Botnet Attack dataset (N-BaIoT) collection that represents nine IoT devices, such as baby monitors, security cameras, doorbells, and thermostats. Each of the nine datasets is classified as either an attack or benign class. These attacks are generated using two botnets. Only three datasets are used, including the Ecobee thermostat, Ennio doorbell and Samsung SNH 1011 N webcam, with 1,566,598 samples, of which 104,363 represent the benign class, while 1,462,235 describe the attack class. For all subdatasets, we split the data into 70% for training and 30% for testing.

Kahani et al. [16] did not use any dataset. Instead, data is generated when testing the implementation to represent the patients’ accounts and health records. Meisami et al. [17] did not use any dataset as well—instead, a discussion of the theoretical aspects of the model, including its security and privacy features.

A detailed summary of the datasets used in the reviewed studies is presented in Table 2. Entries marked as theoretical represent conceptual or survey-based studies that did not employ a specific dataset but contributed analytical frameworks or taxonomies.

## 5. Analytical Models Used

In this section, we will discuss the analytical models used in some of the selected research papers studied for our work, and a tabular summary of the analytical models used in those papers for the survey.

Alotaibi et al. [15] propose a stacked deep learning architecture from five pretrained residual networks (ResNet) for cyberattack detection against IoT devices. Each pretrained ResNet model is made with ten ResNet blocks with two convolutional layers with the same settings assigned. After the ResNet blocks, the data is transferred to the new meta-algorithm, consisting of two dense layers (40 and 20 neurons), followed by softmax to compute the class score. Each of the two convolutional layers comprises 16 convolutional filters that output 16 feature maps, where the activation function ReLu is used. Additionally, average pooling of size two with striding two is used after the last ResNet block to reduce the feature size as it had superior results to max pooling. The authors use cross entropy as the loss function when training and the Adam optimizer because of the advantages of both RMSProp and AdaGrad optimizers.

Kahani et al. [16] propose an authentication and access control manager (AAM) server responsible for both the authentication and access control. Authentication is achieved with the Schnorr zero-knowledge identification protocol, which is used as a challenge-response protocol to authenticate the identity of the healthcare practitioner anonymously. To establish secure communication between multiple actors, a combination of public and private keys generated by the Derive Unique Key per Transaction (DUKPT) scheme is used, which alters the session key each new session to strengthen the security of the communication. Access control is defined by an intention tree, where different tree nodes represent a hierarchical relationship-like structure that represents various healthcare professions. The result of the tree is the minimum rights that a healthcare provider needs to satisfy the data request or will be denied if outside his rights. Firstly, the user registers with the service provider where the ID is generated. Secondly, the user selects a Base Derivation Key (BDKAM) to create a session key and sends it encrypted alongside some key parameters to the AAM, responsible for the authentication and access control, where the AAM responds with an encrypted random number and timestamp. Thirdly, the user combines his ID, the random number and previous key parameters to calculate y, as seen in Equation (1) [16], which is transmitted to AAM.y = ID × e + r × modq(1)

After being authenticated and granted access, the AAM responds with a validation token. Finally, the user uses the token to retrieve the patient’s encrypted data from the cloud server, where the user can decrypt it with the shared key. This protocol is fundamental to the proposed solution of Kahani et al. [16], which manages secured data sharing for e-health services.

Meisami et al. [17] propose a blockchain-based protocol for e-health approaches that does not use a trusted third party and incorporates an efficient privacy-preserving access control method. The proposed model architecture shows four main modules. Wearable IoT devices and patients’ phones are responsible for gathering and temporarily storing patient data. IoT devices typically have low storing space and computational power, while phones have larger storing space and computational speeds and can transmit data through wireless communications. Medical staff are the physicians and nurses who want to download the patient’s data for analysis and treatment. The blockchain module stores the pointers to the data and not the actual data. It also stores access policies and removes the need for a trusted third party. The off-chain storage module is responsible for storing the patient’s encrypted data. It uses an Interplanetary File System for the peer-to-peer distributed file system.

Throughout the protocol, Meisami et al. [17] use three cryptographic functions: the SHA-256 as the hash function, AES for the symmetric key encryption, and ECDSA with a secp256k1 curve for the digital signature algorithm. Initially, a staff m and patient p generate private and public keys to sign and send transactions to the blockchain server and a secret key for the AES encryption. Secondly, the patient registers the data access permissions by assigning policies that indicate what permission message a medical staff has over the patient. These permission policies can later be changed or revoked. After the patient’s data is saved on the off-chain storage, it can be accessed by the medical practitioner by first checking if the staff has permission access. If the staff is granted access permissions, the patient’s encrypted data can be downloaded with the following protocol.

Table 3 summarizes the analytical models and architectures applied in privacy-preserving IoT studies, including structural configurations and evaluation metrics to enhance reproducibility.

## 6. Discussions

The survey findings reveal a clear evolution of privacy-preserving research in Internet of Things (IoT) systems from centralized security frameworks to distributed, intelligent, and energy-aware architectures. This transition is driven by the growing heterogeneity of IoT environments—encompassing edge devices, cloud services, vehicular nodes, and UAV-assisted networks—where static, one-size-fits-all privacy mechanisms are no longer effective.

### 6.1. Interpretation of Findings

Across the analyzed studies [1,2,3,4,5,6,7,8,9,10,11,12,13,14,15,16,17,18,19,20,21,22,23,24,25,26,27,28,29,30,31,32,33,34,35,36,37,38,39,40,41,42,43,44,45,46,47,48,49,50,51,52,53,54,55,56,57,58,59,60,61,62,63], four dominant privacy-preserving paradigms emerge: encryption and blockchain-based techniques, learning-based frameworks (particularly federated learning), edge/cloud hybrid models, and application-specific architectures.

Encryption-based methods continue to serve as the foundation for ensuring confidentiality, integrity, and authentication. Yet, their direct implementation on low-power IoT devices remains impractical due to computational intensity and key-management complexity. Blockchain offers decentralized trust and auditability but introduces latency and consensus overheads that are incompatible with real-time IoT applications.

In contrast, federated learning (FL) and differential privacy (DP) have proven valuable in preserving data locality while maintaining model accuracy. The 2024–2025 surveys [56,57,59,63] confirm that FL reduces data exposure risks; however, it remains susceptible to gradient inversion and model poisoning attacks, especially under non-IID data distributions. Integrating secure aggregation and homomorphic encryption improves privacy but increases energy and communication costs—a trade-off now at the center of IoT privacy research.

Edge and cloud co-design models [54,61,62] address these trade-offs by relocating privacy computation closer to the data source. They offer lower latency and reduced transmission exposure but must reconcile heterogeneous hardware, limited memory, and diverse network conditions. This has spurred interest in Trusted Execution Environments (TEE) and zero-trust architectures for edge nodes.

Finally, application-specific privacy frameworks [55,58,60] emphasize context-dependent protection, particularly for smart homes, healthcare, and vehicular IoT. These domains expose privacy in unique ways: smart homes risk behavioral inference from metadata; e-health faces data sharing and compliance constraints; vehicular IoT demands real-time, low-latency authentication. Collectively, they demonstrate that privacy cannot be generalized—it must be contextualized to device capabilities, connectivity, and user expectations.

A comparative summary of representative studies and their respective strengths and limitations is provided in Table 4.

As highlighted in Table 4 despite significant advancements, each technique family faces distinct challenges in scalability, computation cost, and adaptability—which are analyzed in the following subsections.

The convergence of encryption, federated learning, and blockchain technologies is increasingly viewed as the most promising direction for achieving scalable and verifiable IoT privacy.

The overall interaction among these techniques and their operational layers is depicted in Figure 4, which illustrates how device-level encryption, federated model aggregation at the edge/cloud, and blockchain-based audit trails collectively enable secure and transparent data processing across heterogeneous IoT environments.

### 6.2. Cross-Analysis with Research Questions

The synthesized literature answers the research questions defined in Section 3.1 as follows:RQ1 (Techniques): Privacy in IoT is dominated by four families—encryption, blockchain, FL/DP, and hybrid edge–cloud frameworks—each addressing different parts of the data lifecycle.RQ2 (Devices/Architectures): Most solutions target edge nodes and cloud infrastructure, while ultra-constrained sensor nodes remain under-protected.RQ3 (Threats): Commonly mitigated threats include data leakage, man-in-the-middle attacks, inference attacks in ML, and unauthorized access; emerging threats involve adversarial learning and blockchain data linkage.RQ4 (Comparative Effectiveness): FL and blockchain achieve decentralized privacy but trade off energy, latency, and communication efficiency; lightweight encryption still dominates resource-limited nodes.RQ5 (Gaps): Few studies present unified, end-to-end architectures that balance privacy, scalability, and energy efficiency across heterogeneous IoT tiers.

Building upon the cross-analysis of the reviewed studies and their responses to the formulated research questions, the key unresolved challenges and potential research directions can be synthesized. Table 5 summarizes the identified research gaps and corresponding recommendations extracted from the surveyed literature between 2016 and 2025, highlighting where current privacy-preserving approaches in IoT systems still fall short and what strategies may address these shortcomings in the next generation of designs.

### 6.3. Emerging Trends and Research Implications

Several strong trends and implications arise from the collective analysis:Shift toward Decentralization: Future IoT privacy will rely on federated, peer-to-peer, and blockchain-enabled models rather than centralized authorities.Privacy–Energy Co-Optimization: Energy-aware encryption and adaptive training in FL are emerging to sustain privacy without depleting device resources.Edge Intelligence and Lightweight ML: Integrating TinyML and hierarchical FL supports on-device learning while minimizing data exposure, but model compression and personalization must be improved.Cross-Domain Privacy Frameworks: Unified architectures spanning smart homes, healthcare, and vehicular IoT are required to enable interoperability and standardization.Regulatory Alignment and Ethical Data Governance: The growing enforcement of GDPR-style privacy laws globally demands compliance-by-design models embedded into IoT systems.Hybrid Architectures: Combining blockchain, FL, and differential privacy offers complementary strengths—trust, decentralization, and statistical anonymity—suggesting a clear direction for next-generation IoT privacy frameworks.

### 6.4. Synthesis

Overall, the literature demonstrates rapid maturation of privacy-preserving research from isolated technical mechanisms toward multi-layered, adaptive, and energy-efficient IoT privacy ecosystems. However, the trade-off between privacy strength and performance efficiency remains unresolved. The path forward involves designing Workload- and Context-Aware Privacy Frameworks capable of dynamically balancing computation, communication, and confidentiality in heterogeneous IoT networks.

The cumulative findings from the analyzed studies reveal an evolving research trajectory toward decentralized, adaptive, and energy-efficient privacy frameworks. The strategic path linking these research gaps to long-term objectives is illustrated in Figure 5, which presents a consolidated roadmap for future investigations in privacy-preserving IoT systems.

These insights motivate the future recommendations presented in Section 7, which outline strategic directions for developing robust, scalable, and sustainable privacy-preserving solutions in forthcoming IoT deployments.

## 7. Conclusions and Future Directions

This survey presented a comprehensive analysis of privacy-preserving techniques in Internet of Things (IoT) systems across diverse domains such as healthcare, smart homes, industrial IoT, and vehicular networks. By reviewing peer-reviewed studies and technical reports published between 2016 and 2025, the study systematically classified existing approaches into encryption-based, learning-based, blockchain-enabled, and hybrid edge–cloud frameworks to address five core research questions on applied methods, device layers, privacy threats, comparative performance, and open challenges. The findings reveal a clear shift from centralized, encryption-centric models to decentralized, intelligent, and energy-aware architectures. While classical cryptographic techniques such as AES, DES, and homomorphic encryption provide strong confidentiality, they impose computational and energy burdens on constrained IoT devices. Blockchain technologies enhance integrity and trust through decentralization but introduce latency and power overheads that limit their suitability for real-time applications. Federated learning and differential privacy offer promising solutions for secure data analytics by preserving data locality and enabling collaborative intelligence, yet remain susceptible to gradient inversion, poisoning, and synchronization attacks at scale. Edge and cloud frameworks alleviate part of these constraints by performing computation closer to data sources, thereby reducing exposure and latency, though heterogeneity and the lack of unified security standards continue to hinder interoperability. Across all domains, privacy requirements are shown to be highly contextual—differing for wearable sensors, industrial actuators, and UAV nodes in terms of latency tolerance, computational capacity, and regulatory compliance. Persistent limitations include high computational cost, fragmented interoperability, limited benchmarking datasets, and a lack of real-world validations. To overcome these challenges, future research should prioritize the development of lightweight, energy-aware cryptographic and federated learning schemes; design hybrid frameworks that integrate the complementary benefits of blockchain, federated learning, and differential privacy; employ adaptive edge intelligence through TinyML for dynamic on-device privacy enforcement; and promote standardized cross-domain interoperability and open-source testbeds for benchmarking under realistic deployment conditions. Moreover, embedding privacy-by-design principles in alignment with emerging regulations such as GDPR and HIPAA will be crucial for establishing trustworthy, scalable, and sustainable IoT ecosystems. Overall, IoT privacy research is transitioning toward autonomous, decentralized, and context-aware systems capable of intelligently balancing performance, energy, and confidentiality, providing a solid foundation for the design of next-generation hybrid and workload-aware privacy frameworks.

## Figures and Tables

**Figure 1 sensors-25-06967-f001:**
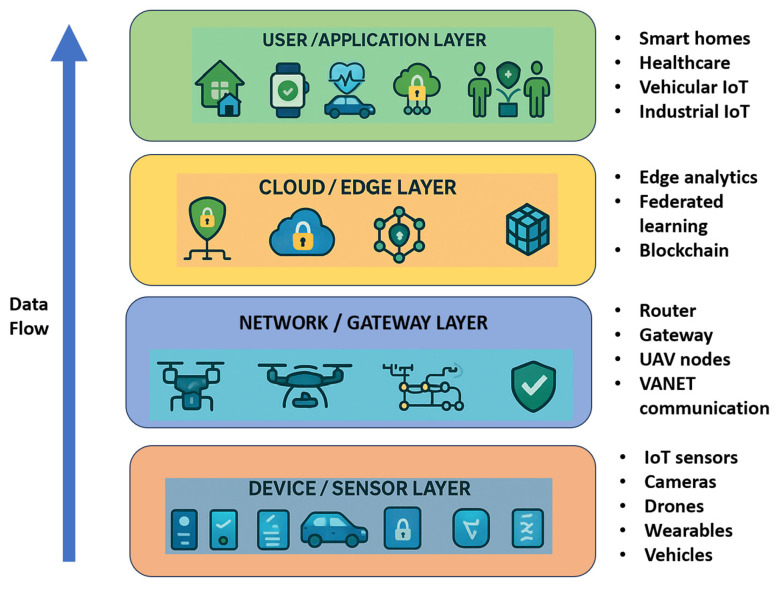
Conceptual overview of privacy preservation in IoT systems showing security challenges and protection mechanisms across different layers of the IoT architecture.

**Figure 2 sensors-25-06967-f002:**
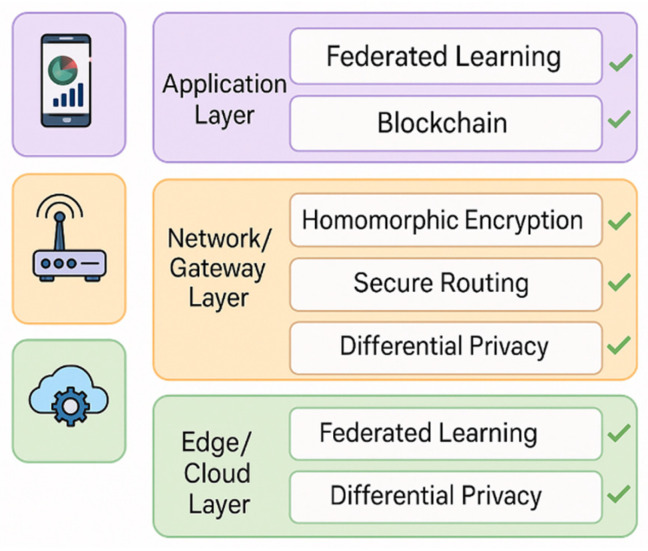
Comparative mapping of major privacy-preserving techniques across IoT layers.

**Figure 3 sensors-25-06967-f003:**
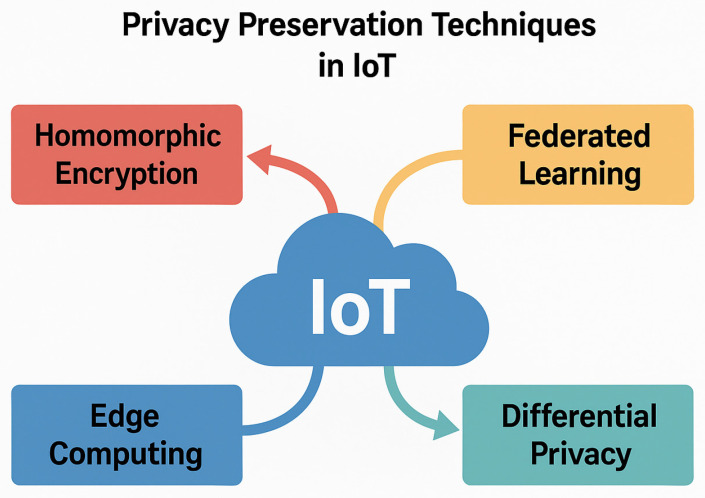
Taxonomy of Privacy-Preserving Techniques in IoT Systems.

**Figure 4 sensors-25-06967-f004:**
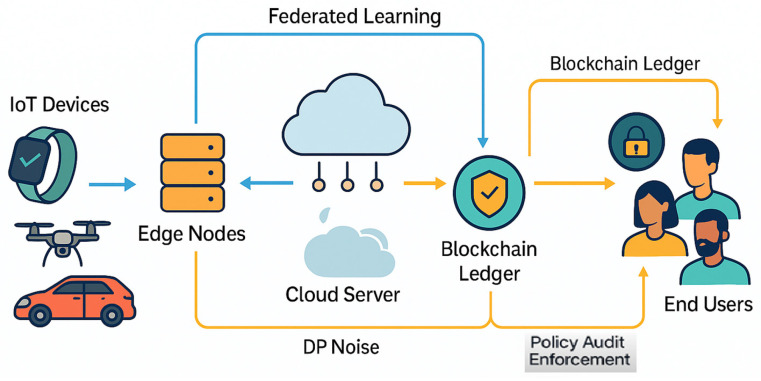
Conceptual hybrid framework integrating encryption, federated learning, and blockchain for end-to-end privacy preservation in IoT environments.

**Figure 5 sensors-25-06967-f005:**
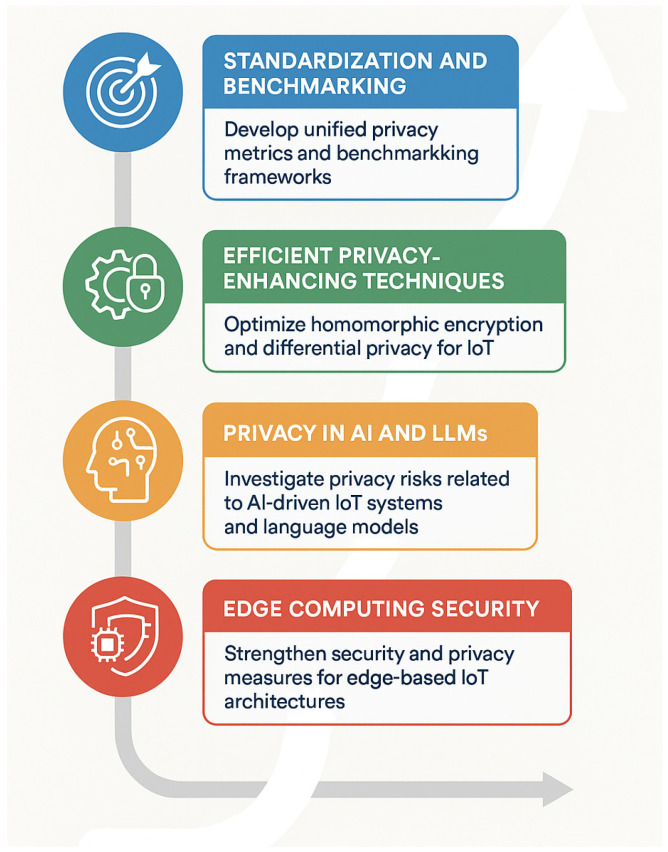
Research gaps and future directions in IoT privacy-preserving systems.

**Table 1 sensors-25-06967-t001:** Summary of privacy-preserving techniques in IoT: categories, methods, datasets, findings, and limitations.

Ref.	Application Domain	Technique Category	Method/Model (Keywords)	Dataset(s)	Key Findings	Noted Limitations
[11]	IoT networks	Learning-based (feature eng.)	Preprocessing + feature selection + J48/Bagging	MQTT-IoT-IDS-2020; NSL-KDD	Up to 99.86% accuracy with proper preprocessing/feature selection	Possible overfitting; limited cross-domain validation
[15]	General IoT (cyberattacks)	Learning-based (DL)	Stacked ResNets + meta-classifier	ICS Cyberattack; N-BaIoT	High accuracy, low per-packet latency	Overfitting risk; limited metrics beyond accuracy
[16]	e-Health cloud	Access control/crypto	AAM + ZK (Schnorr); fine-grained access	Impl. prototype	Confidentiality with manageable latency	Latency under load; scalability vs. stronger crypto
[17]	e-Health (IoT)	Blockchain + access control	AES + ECDSA + SHA-256; on-chain pointers	Conceptual (no dataset)	Decentralized access control; integrity	No empirical overhead analysis
[18]	General IoT	Differential privacy	Event-level DP traffic shaping	Synthetic streams	Privacy–delay trade-offs quantified	Bursty traffic harder to hide efficiently
[19]	e-Health monitoring	Blockchain + IPFS	Re-encryption proxy; PoA chain	Prototype	Secure, scalable storage split (on/off-chain)	Web-style system; real-time path untested
[20]	Fog/Cloud IoT	TEE-based aggregation	Intel SGX; heterogeneous data	Concept/prototype	Privacy for heterogeneous aggregation	Real-world deployment pending
[21]	Vehicular IoT	Blockchain auth.	Smart contracts; hash anchoring	Concept/prototype	Tamper-resistance for vehicle data	Latency/fault tolerance not analyzed
[31]	General IoT	Survey/Taxonomy	Privacy models; data minimization	—	Clear layering of privacy concerns	No implementation/experiments
[33]	Industrial IoT	Blockchain + ML + DP	PriModChain (FL + DP + contracts)	MNIST	Combines trust + privacy in ML sharing	High federation latency
[37]	Large-scale IoT	Lightweight FL	FedL (privacy-preserving)	MNIST	Linear-time growth with users	HE cost; still non-trivial overhead
[40]	Society 5.0 IoT	Context-aware sec.	CASE (post-encryption reduction)	UCI activity	Reduces post-encryption data size	Linear delay increase
[54]	Cross-domain IoT	Survey (encryption/blockchain)	Comprehensive taxonomy	—	Synthesizes device–network–cloud threats	Highlights energy/latency overheads
[55]	Consumer IoT (smart home)	Traffic privacy	Encrypted traffic analysis survey	—	Metadata leakage even without payloads	Need stronger traffic shaping/obfuscation
[56]	Edge IoT	FL + TinyML	On-device FL under constraints	—	Feasible FL/TinyML co-design	Accuracy–energy–latency trade-offs
[57]	Edge-IoT	FL security survey	FL + DP + HE + secure agg.	—	Catalogs FL privacy risks/defenses	Open issues: poisoning, non-IID, comms
[58]	Smart homes	Domain survey	AuthN/AuthZ; lightweight crypto	—	Domain-specific threat landscape	Fragmented device ecosystems
[59]	IoT intrusion detection	FL application	FL-based IDS (privacy-preserving)	Network traces	Preserves data locality; good detection	FL robustness to attacks still open
[60]	General IoT	Cybersecurity survey	Holistic IoT security incl. privacy	—	Broad coverage of threats/controls	High-level; fewer empirical results
[61]	User-centric IoT	Personal Data Stores	Privacy-aware PDS; consent/usage control	—	User control and transparency patterns	Adoption/standardization challenges
[62]	IoT + Cloud	Survey (AI + privacy)	DP, HE, AI-integrated pipelines	—	End-to-end pipeline considerations	Many proposals lack deployment data
[63]	Hierarchical edge IoT	FL personalization	Fed. learning on non-IID hierarchical edges	—	Personalization improves FL quality	Security/privacy of personalization layers

**Table 2 sensors-25-06967-t002:** Datasets used in the reviewed IoT privacy studies, including their domain, size, preprocessing steps, and indication of empirical versus theoretical nature.

Ref.	Dataset/Source	Domain/Application	Size/Samples	Preprocessing and Description	Empirical/Theoretical Note
[3]	Yale B Face Database	Face recognition for IoT cameras	2414 images of 38 subjects	Images resized to 64 × 64 pixels; normalized grayscale; Bloom-filter encoding applied before classification	Empirical study
[7]	Custom IoMT Breast-Cancer Dataset	Healthcare (IoMT)	5200 labeled records	Feature extraction (texture + shape), normalized; trained with CNN/ANN	Empirical study
[10]	Yoga Posture Dataset (Kaggle)	Human-posture sensing via IR sensors	93,200 images of 26 postures; 224 × 224 px resolution	Images annotated manually; balanced per class; lighting and angle normalization performed	Empirical study
[11]	MQTT-IoT-IDS 2020/NSL-KDD	Intrusion detection for IoT networks	~370 k samples (45 features)	Data cleaning, feature scaling, correlation-based feature selection	Empirical study
[13]	–	Industrial IoT privacy framework	N/A	Conceptual framework without dataset; analytical comparison only	Theoretical study (no data)
[15]	ICS Cyberattack/N-BaIoT	Cyberattack detection	100 k network traces	Normalized flow features; applied stacked ResNet for classification	Empirical study
[16]	Prototype logs (hospital IoT)	e-Health access control	8000 transaction logs	AES-encrypted patient data tested in simulated hospital network	Empirical study
[17]	–	e-Health blockchain privacy model	N/A	Architecture diagram only; simulated workflow	Theoretical study
[18]	Synthetic IoT Streams	Differential privacy traffic shaping	1 M events	Laplace noise applied; latency vs. privacy ε measured	Empirical study
[19]	–	Blockchain + IPFS storage for health data	N/A	Framework described; no dataset	Theoretical study
[20]	Edge-Gateway Prototype Logs	Fog/Cloud aggregation	25 k records	Simulated heterogeneous devices; TEE latency measured	Empirical study
[21]	Vehicular IoT Simulation (Veins/SUMO)	VANET blockchain auth.	10 k vehicle events	Cryptographic hash and delay measured under mobility	Empirical study
[33]	MNIST	FL + DP model evaluation	60 k images 28 × 28 px	Data normalized; DP noise added before aggregation	Empirical study
[54]	–	Cross-domain IoT survey	N/A	Review of multi-layer IoT privacy datasets	Theoretical survey
[56]	–	FL + TinyML edge framework	N/A	Conceptual FL prototype; simulation results only	Theoretical/simulation
[57]	–	FL security survey (Edge-IoT)	N/A	Literature synthesis	Theoretical survey
[59]	Network Trace Dataset (CICIDS 2018)	Intrusion detection via FL	~80 k network flows	Flow normalization; feature scaling before FL training	Empirical study
[61]	–	IoT personal data store framework	N/A	Architecture discussion; no datasets	Theoretical study
[62]	–	IoT–Cloud AI privacy survey	N/A	Theoretical integration of AI and cloud privacy	Theoretical survey

**Table 3 sensors-25-06967-t003:** Analytical and architectural models employed in IoT privacy research, detailing their structure, mathematical formulation, and evaluation objectives.

Ref.	Model/Framework	Technique Category	Key Architectural or Mathematical Details (Plain-Text)	Evaluation Metric/Goal
[3]	Hybrid ML Pipeline (Decision Tree + SVM + Naïve Bayes)	Feature-based Learning	Ensemble voting classifier combining probabilistic and margin-based learners; normalized feature vector x in R^45.	Accuracy, Precision, Recall
[7]	CNN + ANN (IoMT Breast-Cancer Diagnosis)	Deep Learning	5 Convolution layers (3 × 3 kernels) + 2 fully connected layers; ReLU activation; Softmax output; dropout rate 0.3.	Accuracy, Loss, F1-Score
[10]	Random Forest + Threshold Sensing	Shallow ML	Feature extraction from IR-sensor frames; temporal smoothing filter applied.	F1 = 0.9989, Precision, Recall
[11]	J48 Decision Tree + Bagging + Feature Selection	Classical ML IDS	Correlation-based Feature Selection (CFS); Bagging ensemble; 10-fold cross validation.	Accuracy 99.86%
[13]	Privacy Index Computation Model	Analytical Model	Privacy Index PI = 1 − (Sp/St), where Sp = sensitive data protected, St = total data collected.	Privacy Index
[15]	Stacked ResNet for Cyberattack Detection	Deep CNN	5 ResNet blocks (each Conv + BatchNorm + ReLU + skip connection); Fusion layer concatenates multi-scale features; Softmax output (12 classes).	Accuracy, ROC-AUC, Latency
[16]	Zero-Knowledge Proof (ZKP) + Access Control	Cryptographic Protocol	Schnorr-based ZKP: gr = a · yc (mod p); AES-128 encryption; ECC for key exchange.	Response Time, Security Level
[17]	Blockchain + Attribute-Based Encryption (ABE)	Hybrid Framework	Smart-contract-controlled ABE with SHA-256 hash indexing and multi-signature verification.	Integrity Ratio, Latency
[18]	Event-Level Differential Privacy Model	Statistical Model	Laplace mechanism: x′ = x + Laplace(Δf/ε); evaluated privacy–delay trade-off.	Mean Latency, ε–Utility Curve
[19]	Blockchain + IPFS Hybrid Storage	Distributed System	Off-chain storage for encrypted payloads; on-chain metadata hashes; Proof-of-Authority consensus.	Access Latency, Throughput
[20]	Trusted Execution Environment (TEE) Aggregator	Secure Hardware Model	Intel SGX enclave executing encrypted aggregation; remote attestation enabled.	Aggregation Delay, Power
[21]	Blockchain-Based Vehicular Auth	Security Protocol	Smart contracts + hash chain; ECDSA elliptic-curve keys for authentication.	Avg. Tx Delay, Packet Loss
[33]	PriModChain (FL + DP + Blockchain)	Hybrid Privacy Framework	FL global model update: w_t = Σ_k (p_k · w_k); DP noise added before aggregation; on-chain update logging.	Accuracy, Latency, Privacy Budget
[37]	Lightweight Federated Learning (FedL)	Distributed Learning	Gradient compression ratio ρ = 0.5; secure aggregation using homomorphic encryption.	Accuracy, Communication Cost
[40]	Context-Aware Security Engine (CASE)	Contextual AI	Feature reduction via PCA; context-triggered encryption selection based on device state.	Accuracy, Response Time
[54]	Multi-Layer IoT Survey Model	Taxonomy Model	Classifies privacy mechanisms by layer (Device, Network, Edge, Cloud).	Conceptual Taxonomy
[56]	FL + TinyML Co-Design Framework	Edge Learning	Quantized 8-bit CNN for microcontrollers; FedAvg algorithm with local epoch E = 5.	Accuracy, Energy Consumption
[57]	Federated Learning Security Survey	Analytical Framework	Comparative analysis of HE, DP, and secure aggregation methods in Edge FL.	Conceptual Synthesis
[59]	FL-Based Intrusion Detection System (IDS)	Federated Application	FL with Adam optimizer; 3 dense layers (64-32-16 neurons); uses CICIDS 2018 dataset.	Detection Rate, F1-Score
[61]	Privacy-Aware Personal Data Store (PDS)	Data Management Model	Semantic ontology-based data schema; policy engine manages access tokens.	Qualitative Evaluation

**Table 4 sensors-25-06967-t004:** Strengths and limitations of selected privacy-preserving approaches in IoT systems, covering encryption, blockchain, federated learning, and hybrid edge–cloud models.

Ref.	Technique/Focus	Major Strengths	Key Limitations
[3]	Bloom-filter-based face recognition (IoT)	Lightweight storage, 92% accuracy on Yale B dataset	Limited to facial data; prone to false positives
[7]	AES + Triple DES for IoMT (breast-cancer detection)	High classification accuracy (CNN 98.5%, ANN 99.2%)	High memory/storage demand; hardware-heavy
[10]	Device-free posture recognition via IR sensors	Near-perfect F1-score (0.9989); privacy-preserving sensing	Low-resolution sensors; deployment-scale untested
[11]	Feature-selection + ML for intrusion detection	Accurate (99.86%); shows preprocessing impact	Possible overfitting; limited datasets
[15]	Stacked ResNet for IoT cyberattack detection	High predictive accuracy; real-time packet analysis	Potential overfitting; lacks performance metrics
[16]	Zero-Knowledge + access control in e-Health	Strong authentication and anonymity	High response latency under load
[17]	Blockchain-based e-Health privacy	Decentralized access control; integrity	Theoretical; no performance validation
[19]	Blockchain + IPFS for medical data	Combines encryption, off-chain storage	Tested only as web prototype
[20]	TEE-based data aggregation	Privacy for heterogeneous data	Unimplemented for real IoT workloads
[21]	Blockchain for vehicular IoT	Secure, decentralized communication	Latency/fault tolerance not studied
[28]	G-BHO meta-heuristic for IIoT privacy	High security gain (>89%)	No real-world validation
[33]	PriModChain (Blockchain + FL + DP)	Integrates trust, privacy, learning	High federation latency; heavy computation
[37]	Lightweight FL (FedL)	Linear scaling with users	Homomorphic encryption cost high
[40]	CASE model for Society 5.0 IoT	Reduces post-encryption data size	Increases network delay linearly
[54]	Survey on IoT privacy/security	Comprehensive taxonomy; identifies multi-layer threats	High-level overview; lacks implementation metrics
[55]	Consumer IoT traffic analysis	Exposes metadata privacy leaks	Needs stronger traffic obfuscation
[56]	FL + TinyML on edge devices	Enables on-device learning under constraints	Energy–accuracy trade-off unresolved
[57]	FL data security survey (Edge-IoT)	Synthesizes FL privacy risks and countermeasures	Communication cost; non-IID challenges
[58]	Smart-home IoT privacy survey	Domain-specific threat insights	Fragmented vendor ecosystems
[59]	FL-based intrusion detection	Preserves data locality; strong detection	FL robustness to poisoning untested
[60]	IoT cybersecurity overview	Broad coverage of IoT threats and controls	Limited empirical depth
[61]	Privacy-aware personal data stores	User-centric transparency and consent	Lacks large-scale adoption examples
[62]	AI-integrated IoT cloud privacy survey	End-to-end data-pipeline view	No deployment metrics; conceptual
[63]	Hierarchical FL personalization	Improves FL accuracy for non-IID data	Security/privacy of personalization layers open

**Table 5 sensors-25-06967-t005:** Summary of identified research gaps and corresponding recommendations across privacy-preserving IoT studies (2016–2025).

Ref.	Identified Research Gap	Recommended Future Direction
[3]	Bloom filter encoding is non-reversible and lacks data backup mechanism.	Introduce reversible key-based encryption integrated with bloom encoding.
[7]	CNN-based IoMT model demands high storage and computing resources.	Employ cloud/edge offloading and lightweight model compression.
[10]	Limited dataset (26 postures) reduces model generalization.	Extend experiments with larger, diverse subject datasets.
[11]	Handling of outliers and unbalanced data not discussed.	Integrate anomaly detection and adaptive re-sampling strategies.
[13]	Comparison with FL and SplitNN models missing.	Evaluate classifier performance vs. communication cost using federated settings.
[15]	Possible overfitting and lack of broader metrics.	Validate on cross-domain datasets and include latency/energy analysis.
[16]	Limited testing of non-common network threats.	Extend experiments to DDoS, poisoning, and replay attacks.
[17]	Model not implemented; computational cost unknown.	Prototype deployment for overhead and scalability analysis.
[18]	Shaper design efficiency under burst traffic uncertain.	Optimize event-level DP models using traffic correlation.
[19]	System limited to web-based prototype.	Develop real-time blockchain-enabled health monitoring testbeds.
[20]	Proof-of-concept lacks real-time validation.	Implement full-scale deployment using Intel SGX or AMD SEV.
[21]	Missing comparison for latency and fault tolerance.	Benchmark blockchain-based vehicular IoT under dynamic mobility.
[23]	Limited flexibility and latency in private FL.	Integrate RFID-based automatic on-chain identification.
[25]	Narrow application scope (six IoT use cases).	Broaden dataset; adopt chatbot-based PbD assistance.
[26]	Prototype not tested on real data.	Apply to large-scale finance or insurance IoT systems.
[28]	Evaluated only via simulation.	Validate industrial implementation using live sensor data.
[30]	Equilibrium model theoretical only.	Apply to aggregated IoT datasets for empirical confirmation.
[31]	No implementation evaluation.	Compare privacy-minimization methods experimentally.
[32]	Legal/ethical aspects missing.	Collaborate with healthcare regulators for compliance studies.
[33]	High latency in federated rounds.	Optimize communication scheduling and adaptive intervals.
[35]	Computationally heavy recommender model.	Explore pruning and parallel training to reduce overhead.
[37]	Limited balance between privacy and speed.	Investigate hybrid HE + DP approaches for efficiency.
[39]	Metrics for privacy leakage unclear.	Extend model evaluation with multiple datasets and leak indices.
[40]	Performance comparison incomplete.	Analyze accuracy–latency trade-offs with additional models.
[41]	Extra hashing adds overhead on WSNs.	Develop lightweight hash variants for constrained sensors.
[54]	Lack of empirical validation across multi-layer IoT.	Conduct quantitative benchmarks and real deployments.
[55]	Metadata-level leaks not fully addressed.	Apply advanced traffic obfuscation and packet padding.
[56]	Energy–accuracy trade-off unresolved in FL/TinyML.	Co-optimize energy consumption and model precision.
[57]	Limited exploration of poisoning and communication costs.	Employ secure aggregation and adaptive client selection.
[58]	Fragmented device ecosystems in smart homes.	Standardize protocols and unify vendor authentication layers.
[59]	FL intrusion detection untested against adversarial attacks.	Include adversarial robustness evaluation.
[60]	High-level survey, lacks quantitative depth.	Incorporate performance benchmarking of cited solutions.
[61]	PDS adoption slow due to missing standards.	Establish interoperability frameworks and open APIs.
[62]	Conceptual AI–IoT integration lacks deployment results.	Build AI-driven, privacy-aware IoT cloud demonstrators.
[63]	Personalization layer in hierarchical FL unverified.	Investigate secure personalization preserving user privacy.

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
