# Peer review of "A Survey on Privacy Preservation Techniques in IoT Systems"

_sensors, 2025, doi:10.3390/s25226967_

Round 1
Reviewer 1 Report
Comments and Suggestions for Authors
The paper systematically catalogs IoT privacy techniques but requires enhancements in methodological rigor, technical granularity, and formatting. Post-revision, it could serve as a robust reference for future research. The following issues should be addressed:
- Critical technical descriptions remain superficial (e.g., the operational mechanism of the "event-level differential privacy model" in [18] is not elaborated). The authors should add technical implementation details for pivotal methods (e.g., blockchain in healthcare data sharing) and categorize comparative analyses of strengths/weaknesses (e.g., encryption efficiency, communication overhead in federated learning).
- Table 3 summarizes datasets but exhibits formatting inconsistencies (e.g., [14] labeled as "NA" without explanation). Furthermore, theoretical works like [17] and [19] lack empirical validation, weakening their persuasiveness. The authors should specify dataset sizes, sources, preprocessing steps, and explicitly state the limitations of purely theoretical studies.
- Table 4 lists models but inadequately explains complex architectures (e.g., [15]'s stacked ResNet: layer configurations, fusion strategies). It's better to include architectural diagrams or mathematical formulations to improve reproducibility (e.g., ResNet block parameters, voting mechanisms for ensemble models).
- The discussion of [11]'s feature selection is thorough, but Table 9 oversimplifies limitations (e.g., [10] mentions "low-resolution sensors" without addressing deployment challenges). The authors should propose concrete future directions, such as optimizing lightweight homomorphic encryption for edge devices or designing cross-domain privacy-preserving frameworks.
- Expand the literature review section with more practical IoT scenarios, vanets, and smart homes to include privacy-preserving mechanisms tailored to UAVs and vehicular IoT (e.g. secure and efficient authenticated key management scheme for uav-assisted infrastructure-less iovs, designing secure blockchain-based authentication and key management mechanism for Internet of drones applications, secure remote user authenticated key establishment protocol for smart home environment.
- Grammatical issues occur (e.g., misuse of "where" in "blockchain technology where the decentralization principle is applied"). Section numbering is inconsistent (e.g., subsection 5.1 lacks 5.2). Authors should use professional editing for syntax/structure and standardized section labels.
- "The dataset has 93,200 posture images of 26 postures." (p. 6, lines 180-181). The statement omits resolution, annotation protocols, and bias checks (e.g., diversity of subjects, lighting conditions).
Reviewer 2 Report
Comments and Suggestions for Authors
1) All acronyms must be written in full the first time they appear within text.
2) The quality of English used throughout this paper must be greatly improved.
3) This being a survey paper, you must give a summary of the key findings towards the end of the abstract.
4) The clarity of all figures must be enhanced.
5) Rewrite the introduction section so tat the research area and problem domain are well articulated.
6) Towards the end of the introduction section, articulate the major scientific contributions of this work (preferably in point/bullet form).
7) This paper lacks a clear methodology. You state that :All the research papers that have been used in our survey were published between 2016 to 2023. The papers have been published on various platforms like IEEE, Springer Link, MDPI, IET, ELSEVIER, and ACM."
i) Why have you only considered 2016 to 2023 and yet we are now in the year 2025?
ii) Do you imply that in the years 2024 and 2025 no research works were published in this research domain?
8) You need to introduce the 'Methodology' section and clearly articulate the following:
i) Search String
ii) Data sources (including the links to these data sources, e.g. IEEE Xplore (ieeexplore.ieee.org))
iii)Screening Process
iv) Inclusion and Exclusion Criteria
v) Study Selection Process
vi)Quality Assessment Criteria
9)In the Related Work section, the literature should be arranged in thematic areas. Then, use tables and figures/diagrams to summarize the key findings in each of these thematic areas.
10) What is the need for Data Sources and Types in Section 3? What is the need for Table 3?
11) In the abstract, you indicated that this is a survey paper. However, in Figure 9, you are describing 'Proposed model'
12) What is the need for Figure 9-13?
13) The Discussions is Section 5 is not well done. You were expected to discuss the major findings in this section and offer appropriate interpretations of these findings. However, you have just continued with the literature review of some of the existing works. Why go into detailed descriptions of existing works ( by including Tables and Figures) when you have already provided reference where readers can find this information?
14) The conclusion section has failed to bring out key elements such as a summary of the findings, limitations of the current work, and any future research directions.
15) Revise the list of references to include research works published in the year 2025.
This paper needs extensive revisions in terms of the quality of English language used.
Round 2
Reviewer 1 Report
Comments and Suggestions for Authors
The author have addressed all my concerns, this paper can be accepted.
Author Response
We sincerely thank the reviewer for the positive feedback and kind recommendation. We truly appreciate the time and effort invested in reviewing our manuscript and are grateful that the revisions have met your expectations. We have carefully verified the final version and are uploading the final revised manuscript accordingly.
Thank you once again for your thoughtful review and support toward the acceptance of our paper.
Kind regards,
Rupinder Kaur, Tiago Rodrigues, Nourin Kadir, and Rasha Kashef
Department of Electrical, Computer, and Biomedical Engineering
Toronto Metropolitan University, Canada
Reviewer 2 Report
Comments and Suggestions for Authors
Thank you so much for the great efforts made towards addressing the previous concerns. However, take note of the following minor issues:
1) The conclusion section should be written as one continuous paragraph
2) Avoid references to tables and figures from within the abstract. As such, Table 5 and Figure 5 must be shifted to relevant sections within the paper.
